# Validation of the Curiosity and Exploration Inventory-II in Spanish University Students

**DOI:** 10.3390/healthcare11081128

**Published:** 2023-04-14

**Authors:** Isabel Tarilonte-Castaño, Diego Díaz-Milanés, Montserrat Andrés-Villas, Zaira Morales-Domínguez, Pedro Juan Pérez-Moreno

**Affiliations:** 1Department of Clinical and Experimental Psychology, University of Huelva, 21007 Huelva, Spain; isabel.tarilonte@alu.uhu.es (I.T.-C.); zaira.morales@dpsi.uhu.es (Z.M.-D.); pedro.perez@dpsi.uhu.es (P.J.P.-M.); 2Department of Psychology, Universidad Loyola Andalucía, 41704 Sevilla, Spain; 3Department of Social, Developmental and Educational Psychology, University of Huelva, 21007 Huelva, Spain; montserrat.andres@dpsi.uhu.es

**Keywords:** curiosity, exploration, psychometrics properties, factorial invariance, Spanish

## Abstract

Background: This study aimed to analyse the psychometric properties of the Curiosity and Exploration Inventory-II (CEI-II) to provide evidence of validity for its use in research on health promotion and the quality of life of young Spanish university students. Method: A sample of 807 participants (75.09% female) aged 18–26 years (M = 20.68; SD = 2.13) completed the CEI-II and health and quality of life measures questionnaire. Results: A unidimensional structure was confirmed, but the original two-dimensional structure also showed an adequate fit. The measures obtained from the CEI-II were gender- and age-invariant, which exhibited adequate internal consistency for both the full scale and subscales, and showed a statistically significant relationship with life satisfaction, sense of coherence, and psychological distress. Conclusions: The CEI-II can be used as unidimensional, which is recommended, but also as a two-dimensional measure. Both structures provide reliable, valid, and invariant measures across gender and age of exploratory behaviours in Spanish university students. Furthermore, the results confirm the association between exploratory behaviours and greater health management.

## 1. Introduction

According to the Ottawa Charter for Health Promotion [1], health promotion is the result of factors such as care for oneself and others, the ability to make decisions, and the possibilities that society offers its citizens. Health is created and lived within the framework of everyday life, including education centres, work, and recreation. Thus, these areas are considered to play an essential role in health promotion, which can be achieved by involving the relevant organisations or institutions in the quest to work on improving health-related outcomes [2].

This idea, developed in documents that emerged from the International Conferences on Health Promotion in Ottawa [1], Sundsvall [3], Jakarta, [4] and Mexico [5], generated the lines of action taken by universities in order to promote health, as stated in the Edmonton Charter [6]. This charter states that universities are fundamental agents of health in the community sphere since they provide the context in which educational activities that focus on healthy lifestyles can be developed.

Thus, research has been conducted to clarify the role of various phenomena associated with the health of young people in universities. For example, in Spain, there are data on health-related habits such as adherence to the Mediterranean diet, the practice of physical activity, alcohol consumption, and leisure activities [7,8,9,10,11,12]. However, also, much of this research has focused on the relationship between psychological variables and the university students’ health, as well as the psychometric analysis of the instruments for the measurement of such constructs, in response to the necessity of having valid tools that can identify characteristics related to the wellbeing and health of a specific population. The explored psychological variables that can be found are psychological wellbeing, life satisfaction, sense of coherence, emotional intelligence, resilience, and curiosity, among others [13,14,15,16,17,18].

Curiosity has been a particular focus of attention in studies within the field of Positive Psychology [19]. It has been linked to other constructs, such as creativity or personal creative identity [20], as well as variables associated with education [21] and work contexts [22]. It should also be noted that in numerous studies, curiosity has been related to mental health indicators such as life satisfaction [23], resilience [24], self-efficacy, self-esteem [25] and psychological wellbeing [15,16,26]. In a similar vein, Gallagher and Lopez [15] propose that curiosity can be an indicator of psychological wellbeing, which is a key determinant of an individual’s general health [27]. However, Kashdan et al. [28] indicate that high levels of curiosity could be a risk factor in combination with high social anxiety because they reported greater difficulties managing difficult emotions and hostile impulses, fewer social resources, and less psychological flexibility, which predicts a lack of recognition of the activities as threats and rewards and generates greater risk-taking behaviours than their peers.

There are various instruments for measuring curiosity, such as the State Trait Curiosity Inventory [29], which approaches this construct from an emotional and motivational standpoint, and the Epistemic Curiosity Scale [30], which focuses on intellectually challenging activities. Further, taking Berlyne’s model [31] as a basis, Kashdan, Rose, and Fincham [32] designed the Curiosity and Exploration Inventory (CEI-I) and later the Curiosity and Exploration-II (CEI-II) [33], both of which are regarded as useful evaluation scales for laboratory and survey-based research. Kashdan et al. [32] define a curious person as more likely to recognise, pursue, and absorb new and challenging experiences.

The CEI-II version attempts to reflect the totality of this concept [33] and consists of 10 Likert-type items divided into two subscales of five items each: the stretching subscale, or the motivation of the individual to seek experiences and information from stimuli that involve novelty and complexity, and the embracing subscale, which describes a person’s willingness to manage the novelty and uncertainty of situations.

In recent years, a number of published studies have analysed the psychometric properties of this questionnaire in different cultural contexts [19,34,35,36], studying its internal structure. For instance, the results of the confirmatory factor analysis (CFA) reported by Acun et al. [19] in a Turkish sample indicate the existence of two dimensions. The same is shown in the Indonesian version [35]. Balgiu [36], however, in a Romanian sample, tested the fit for three confirmatory models (with one, two, and three factors, respectively) and found that the data showed an adequate fit for the three-component model where the third factor is obtained by dividing the embracing subscale into two different subscales, although the initial exploratory factor analysis (EFA) presented important cross-loadings of items. In contrast, the study by Ye et al. [34], conducted on the Hong Kong student population, found CFA results that are consistent with a one-dimensional model. Thus, the results of these analyses do not provide consistent evidence regarding the dimensionality of the instrument [34,36].

With regard to the psychometric analysis of the CEI-II [33] in the Spanish population, no studies have been carried out to date. Indeed, a systematic search of the descriptors (“curiosity” or “CEI-II” or “Curiosity and Exploration Inventory” and “psychometr*” or “validit*”) conducted in ProQuest (including APA PsycInfo, APA PsycArticles, PSICODOC, MEDLINE, Psychology and Behavioral Sciences Collection) in October 2022 did not yield any results.

Therefore, the present study aimed to analyse the psychometric properties of the Curiosity and Exploration Inventory-II (CEI-II) in a sample of young Spanish university students. To this end, evidence of its validity on the basis of both its internal structure and its relationships with other variables is provided, as well as an estimation of its reliability based on its internal consistency.

## 2. Materials and Methods

### 2.1. Participants

Stratified random cluster sampling was applied to recruit the participants. The strata (with proportional allocation) were the areas of knowledge. As clusters, first and third-year subjects were randomly selected until completing the quota established for each area of knowledge. To be eligible for participation in the study, individuals needed to be currently enrolled in a degree program at the University of Huelva and explicitly consent to the processing of their data. Excluded from the study were students who were studying abroad (Erasmus) and minors who were under 18 years of age.

A total of 970 students agreed to participate in the study, but some were excluded for various reasons. A total of 31 did not complete the informed consent form satisfactorily, 59 were minors, did not provide their age, or were identified as outliers based on their age (with extreme values greater than 26 years according to the stem and leaf graph), and 73 did not complete all of the study’s instruments or items. Therefore, the final sample consisted of 807 students, of whom 75.09% were female and 24.91% were male, with an age range of 18 to 26 years (M = 20.68; SD = 2.13). The participants in the study came from various areas of knowledge: Arts and Humanities (5.95% of the sample), Engineering and Architecture (1.86%), Natural Sciences (2.23%), Health Sciences (41.88%), and Social and Legal Sciences, which accounted for nearly half of the sample (48.08%). An amount of 48.01% belong to the first academic year, followed by 30.27% in the third, 19.98% in the second, and 1.74% in the fourth. Furthermore, 61.54% declared to move to the city to develop their university studies, while 38.46% did not.

Using the R [37] procedure for creating random samples with quotas by gender and age groups of approximately 50% of cases, the participants were divided into two groups in order to perform an EFA with the first group and CFA with the second. Sub-sample 1 comprised 402 subjects and Sub-sample 2 had 405 subjects, with no statistically significant differences between both groups in terms of gender (χ^2^(1) < 0.001, *p* = 1) or age (t(804.98) < 0.001, *p* = 1).

### 2.2. Instruments

The participants completed a questionnaire on various health-related issues. The questions of the CEI-II were selected, in addition to other related variables of interest to this study.

Curiosity and Exploration. The Curiosity and Exploration Inventory (CEI-II) is a questionnaire developed by Kashdan et al. [33], which measures curiosity using a 10-item Likert-type scale with five response options (ranging from 1 = “Very little or none”, to 5 = “A lot”). The reliability of the scale, as estimated by Cronbach’s Alpha Coefficient, ranges from 0.83 to 0.86 [33]. The Spanish version was adapted through a translation/back-translation procedure carried out by two bilingual translators and supervised by the Spanish team of the WHO Regional Office for Europe’s collaboration study of Health Behavior in School-Aged Children [38]. Once the semantic equivalence was determined between the original and back-translated versions, any differences were debated and amended until agreement was obtained. This procedure was repeated until there were no longer any discrepancies between the meanings of the translated and back-translated scales [39].

Psychological Distress. This was measured using the 12-item version of the General Health Questionnaire (GHQ-12) [40]. The responses range from 0 = “No, not at all” to 3 = “Much more than usual”. A higher score indicates greater levels of psychological distress. Studies performed across 15 countries estimated a high validity coefficients [41]. Furthermore, in Spain some studies found reliability scores from 0.76 to 0.90 [42,43]. The Cronbach’s alpha coefficient obtained in the present study was 0.87.

Life Satisfaction. The Satisfaction with Life Scale (SWLS) [44] was used in its Spanish version [45]. This has five items with response options ranging from 1 to 5, where 1 = “totally disagree” to 5 = “totally agree”. This questionnaire has shown adequate psychometric properties [45]. The Cronbach’s alpha coefficient obtained in the present was 0.84.

Sense of coherence. The Sense of Coherence Scale (SOC-13) created by Antonovsky [46] and extracted from the HBSC study in Spain [38]. The brief version was used, composed of 13 items with 7 Likert type response options that study the frequency with which a person lives certain experiences. The instrument can be used divided into three factors (meaningfulness, comprehensibility, and manageability) or as a single factor. Several research [47] have demonstrated evidence of the validity of this instrument and good internal consistency, with scores ranging from 0.70 to 0.92 [48]. The Cronbach’s Alpha Coefficient obtained in the present study was 0.80.

### 2.3. Procedure

The present research is part of the Health Behavior in University (HBU) study, which is based on a cross-sectional survey design. The data were collected in two different periods during the 2018/19 academic year.

A list of each degree program was gathered prior to the collection of these data and the approximate number of students enrolled in each topic was calculated using information from the previous years. Then, quotas were set in accordance with the area of knowledge, and the degrees and topics within each quota were randomly selected in order to administer the questionnaire.

Data collection was conducted in person by administering the paper questionnaire during one teaching hour. Researchers who had been adequately trained in data collection requested the participation of the students and explained the nature of their participation. It was emphasised that participation was voluntary and that the study was not related to the subjects that they were studying for their degree. They also explained to the participants their right to withdraw from the study and the possibility of leaving any questions blank if they wished. The average time taken to complete the questionnaire was approximately 20 min.

Written consent was obtained from all participants. The current study followed the basic ethical principles of the Declaration of Helsinki and was authorised by Research Ethics Committee of Huelva Centers (CEI) of the Junta de Andalucía (0846-N-19/P1027/19).

The data were computerised using an Excel template that prevented the entry of out-of-range responses.

### 2.4. Data Analysis

The software used were FACTOR [49], to perform the EFA, and R (Version 4.1.3) [37] for data processing and the rest of analyses, by implementing the packages: psych (Version 2.2.3), MVN (Version 5.9), psychometric (Version 2.3), psycho (Version 0.6.1), lavaan (Version 0.6.11.1676), semPlot (Version 1.1.5) and semTools (Version 0.5.6). A polychoric correlation matrix was selected for reliability and factorial analysis calculation.

First, exploratory factor analysis (EFA) was conducted with Sub-sample 1 by using the Diagonally Weighted Least Squares (DWLS). The number of dimensions to be extracted was calculated with the Optimal Coordinates, Acceleration Factor, and Parallel Analysis methods. The mode and the quality of the indicators showed the number of factors. Additionally, a one-dimensionality assessment based on Unidimensional Congruence (UniCo), Explained Common Variance (ECV) and Mean of Item Residual Absolute Loadings (MIREAL) was performed.

Second, confirmatory factor analysis (CFA) was performed with Sub-sample 2. The fit measures applied were the χ^2^ statistic, the chi-square ratio (χ^2^/df), the comparative fit index (CFI), the Tucker-Lewis index (TLI), the root mean square error of approximation (RMSEA), and the standardised root mean square residual (SRMR). Cut-off criteria were as follows: ≤3 for the χ^2^/df ratio; ≥0.90 for the CFI and TLI; ≤0.08 for the RMSEA and SRMR [50].

Third, invariance factor analysis across gender and age was performed in the total of the sample by also using DWLS. In the case of age, four different intervals were made in order to generate similar ordinal categories with a large enough number of participants to carry out the invariance factor analysis across groups. Invariance indicators were a −0.01 change in CFI, paired with changes in RMSEA of +0.015 concerning the least restrictive model [51]. Progressive invariance was tested for four models (configural, metric, scalar, and strict).

Fourth, internal consistency analyses were performed through Ordinal alpha, Guttman split-half, McDonald’s omega and the correlations between the score of each of the items and the total score of the scale and sub-scales were analysed for the global and segmented sample by gender and age group.

Finally, to analyse the evidence of external validity referring to the relationship of the scale score with other variables, the Pearson correlation coefficient was used to describe the association between CEI-II score and GHQ-12, SWLS and SOC-13.

## 3. Results

### 3.1. Descriptive Statistics

The original and the Spanish translation of the items can be found at the Appendix A. Table 1 present the means, standard deviations, skewness, and kurtosis of the 10-items scale. Although the items showed a univariate skew and kurtosis in the range of −1 and +1, the tests of multivariate skewness (χ^2^(220) = 658.401, *p* < 0.001) and multivariate kurtosis (z = 21.419, *p* < 0.001) were both statistically significant, indicating that the data did not follow a multivariate normal distribution.

### 3.2. Evidence of Validity Based on the Internal Structure

An EFA was conducted using the Diagonally Weighted Least Squares (DWLS) extraction method on a polychoric correlation obtained from Sub-sample 1 due to the multivariate non-normality of the items.

First, in order to establish whether the correlation matrix was suitable for conducting the EFA, the results of Bartlett’s sphericity test were assessed, which yielded a statistically significant result (χ^2^(45, *n* = 402) = 1855.733, *p* < 0.001). Likewise, the value of the Kaiser-Meyer-Olkin index for sampling adequacy was 0.89 and the correlation matrix determinant was 0.009, which can be considered adequate.

The different algorithms used to assess the number of factors/dimensions to isolate supported the one factor solution. Nevertheless, we decided to compare the original two-dimensions (2-D) structure of the scale based on Promax rotation with the one-dimension (1-D) structure (Table 2). The one-dimension proposal explained 51% of the variance and showed an CFI = 0.985 and RMSEA = 0.066 (90% CI [0.04, 0.073]) as exploratory estimates, while the two-dimension model explained the 60.57% (Factor 1 = 50.9%; Factor 2 = 9.6%) of the variance, and the exploratory indicators CFI and RMSEA were 0.992 and 0.057 (90% CI [0.035, 0.066]), respectively. However, even when the indexes of two-dimensional model showed slightly better fit, the UniCo, ECV and MIREAL suggest the data can be treated as essentially unidimensional, with values of 0.985, 0.883 and 0.212, respectively.

Regarding the communalities of the items, all values exceeded 0.20, with the lowest being 0.217 and 0.254, respectively, for one-dimensional and two-dimensional models, in the case of the item “1. In novel situations I actively seek as much information as I can”.

Therefore, a CFA may be useful to decide on the model due to the similarities found.

CFA was performed to test these two different models by using Sub-sample 2. The main results are shown in Table 3. Both models showed an adequate Goodness of Fit with similar values in their indexes. Standardised weights for both models are shown in Figure 1. It is necessary to highlight the high correlation (r = 0.97) between the two factors of the two-dimensional model.

Once the adequacy of both models was confirmed, an analysis of gender and age group invariance was performed in each of them. For that purpose, we used the full sample. As shown in Table 4, there was no decrease in CFI lower than −0.01 and an increase in RMSEA greater than 0.015, so the measurement invariance was confirmed in all subsamples. Therefore, both models achieved a strict level of invariance regardless of gender and age differences.

### 3.3. Evidence of Reliability Regarding Internal Consistency

All items have corrected Item-Total correlations ranging from 0.46 (“In novel situations I actively seek as much information as I can”) to 0.76 (“I often look for opportunities to challenge myself and grow as a person”).

The 10-items scale showed an ordinal alpha of 0.91 (95% Confidence Interval: 0.79–0.97), while Cronbach’s alpha for the same items was 0.89, indicating an attenuation effect of −2.28%. This happened with the two-dimensional model, where the factor Stretching and Embracing showed an ordinal alpha of 0.86 (95% Confidence Interval: 0.47–0.98) and 0.83 (95% Confidence Interval: 0.36–0.98), respectively, while their Cronbach’s alpha was 0.83 and 0.79 (attenuation effect of −3.36% and −3.96%). In Stretching, the ordinal alpha would increase by 0.02 if Item 1 was removed, while, in Embracing, the total of the scale the removal of any of the item did not improve their internal consistency.

In addition, the full scale showed a McDonald’s omega of 0.91 while the two-dimensions of the original model showed 0.85 for Stretching and 0.83 for Embracing.

Then, the scale was randomly split into two halves, resulting in a Guttman Coefficient value on average of 0.91, with a minimum of 0.86 and a maximum of 0.93, for the 10-items scale. Regarding the sub-scales from the two-dimensional model, Stretching shown a Guttman Coefficient on average of 0.80 (min: 0.77; max: 0.83) and Embracing a 0.78 (min: 0.77; max: 0.79).

Furthermore, the scale and sub-scales showed a good internal coefficient by gender and age group (Table 5), except for 22–23 years old group with a McDonald’s omega and Guttman coefficient lower than 0.75 for the Embracing dimension.

### 3.4. Evidence of External Validity Regarding the Relationship with Other Variables

Criterion validity was also explored. To this aim, a Pearson’s correlation matrix between the CEI-II total score and subscales and other theoretically related instruments was created (see Table 6). Correlation coefficient values indicate positive, small (r ≥ 0.10 and r ≤ 0.30) and significant relationships (*p* < 0.001) of CEI-II total score and both subscales with SWLS and SOC-13 score, while the relationship with the GHQ-12 score was negative. Finally, the correlation between both CEI-II subscales was high (r > 0.50, *p* < 0.001).

## 4. Discussion

The Curiosity and Exploration Inventory-II was created as a measurement instrument to evaluate exploratory behaviours related to curiosity based on Berlyne’s model [31]. This instrument has been employed in health promotion studies with samples of university students from various countries [19,33,34,35,36] and has shown good psychometric properties. However, no studies have been found that provide evidence of the validity of the Spanish version of this scale. Thus, given this lack in the literature, the aim of the present study was to provide evidence of its validity and estimate the reliability of the Spanish version of the CEI-II in a sample of university students.

The items of the original scale were reduced by means of EFA, and it was found that the resulting scale had a unidimensional structure. The existence of this structure was supported by CFA and was found to be invariant across gender and age. The instrument also showed a good reliability based on its internal consistency indexes (Ordinal Alpha, McDonald’s Omega and split-half Gutman coefficients) and meaningful correlations with relevant constructs based on wellbeing and health related variables. These results show that the current version of the CEI-II is a valid and reliable measure for its use in Spanish university students, in line with the findings obtained with a sample of Chinese university students [34].

Conversely, a two-dimensional structure was tested with the same procedure, forcing the exploratory factor analysis to a bifactor model, obtaining the same structure found by Kashdan et al. [33]. This two-dimensional structure also showed a good fit according to the CFA, good reliability indexes for the total sample, and was invariant across gender and age. This bifactorial structure differs from the Indonesian [35] and Turkish undergraduates student samples [19], where the item fourth loads in the first factor (Stretching).

Nevertheless, the factors in the two-dimensional model are strongly correlated in the present study (r = 0.97), larger than r = 0.79 and r = 0.85 reported by Kashdan et al. [33]. Regarding previous validation studies of the instrument, none of them reported the matrix that they used, Pearson product-moment correlation was assumed due to the version software used in most of the cases. The assumption of multivariate normal distribution of items was only tested by Ye et al. [34], who used as an extraction method the robust maximum likelihood (RML) estimation due to the violation of the assumption, while the rest of the study used maximum likelihood (ML) method with no assumption check reported. This fact could explain the generation of two-dimensional or superior order factor structures because results may be biased and lead to erroneous decisions regarding the model tested [52]. Lloret-Segura et al. [53] indicate that even though the multivariate normality is violated if a univariate distribution is proximal to the normality and the number of options is five or more, a ML method can be applied. However, they also indicate there is not a complete agreement among authors about the range of skewness and kurtosis to consider an item normally distributed, with ranges from [−1;1] to [−2;2] [54,55,56,57]. This point is crucial because the skewness of the items in both directions, among other elements, can increase the likelihood that the estimators yield inadequate estimates and standard errors [53,54,58], which could explain the differences among solutions obtained in previous works. DWLS provides more accurate parameter estimates in those cases, and the model’s fit is more robust to variable type and non-normality [52].

Despite these methodological and statistical components, several variables of diverse nature could lead to differences in factor structures, ranging from the gender of the participants [59] and ethnic and cultural factors [51,60] to the translations used [61] or even the reading comprehension level of the participants in the sample [62]. In addition, it would be useful to conduct a theoretical review and gather expert advice on the content of the curiosity construct and its internal relations [61].

Furthermore, the present study does not attempt to replicate the structure obtained from the Romanian sample [36] because it was based on principal components analysis (PCA) and not factor analysis, and the solutions obtained by the PCA showed strong cross-loadings among the three factors proposed. Therefore, the results reported do not achieve robust evidence of cross-validation between the PCA and CFA models.

In addition to the methodological criteria, when assessing the studies that preceded the CEI-II, the same authors consider curiosity a personality trait that should be treated as a complete construct, and that was related to positive affective and motivational qualities, among other constructs [63]. Being the development of the CEI-I [32] and CEI-II [33], the ones that indicate the segregation in different factors based on the statistical results but are not totally supported by theoretical reasons.

We found that the CEI-II has adequate reliability for both the full scale and the stretching and embracing subscales. These estimates were made with Ordinal Alpha, McDonald’s coefficient and by employing the Split-Half method (by randomly splitting the scale items and each subscale). Various procedures were used to establish the extent to which the results depended on the method used. In most cases, results were obtained indicating good reliability (greater than 0.75). Only the Embracing subscale showed some coefficients under the proposed limit of 0.75 regarding the 22–23 age group, but none of them was inferior to 0.70. Therefore, it could still be considered reliable for that range of age. In addition, the attenuation effect was close to 2% for the total scale and between 3% and 4% for each subscale indicating small variations regarding Cronbach’s alpha coefficient, which should be considered in future applications of the instrument. These findings indicate CEI-II in its Spanish version is a consistent measure of curiosity, despite the low number of items in the instrument (10 in the case of the full scale and five in the case of each subscale).

If an item is removed, alpha values were obtained that could indicate the poor suitability of certain items. Specifically, in the stretching subscale, removing the item “In novel situations I actively seek as much information as I can” could increase the reliability of the scale. However, this increase maintained the value of the coefficient within the 95% confidence interval of the reliability estimate and thus did not significantly affect the psychometric indicators, which could lead to an under-representation of the contents expressed in the item.

Finally, the correlations with psychological distress, life satisfaction, and sense of coherence were shown to be statistically significant with the global score and both subscales. Specifically, a negative correlation with psychological distress based on GHQ-12 has been observed. This is due to the fact that higher scores on this instrument are taken to indicate poorer mental health [42,43]. Therefore, a person with a higher degree of curiosity will obtain a better score on psychological wellbeing along with a tendency to promote positive subjective experiences [32] and to manifest their ability to adapt to changing environments in order to pursue their goals [33], which is key in the improvement and preservation of health in line with the salutogenic model [46,64,65]. In the same vein, the positive relationships found with life satisfaction suggest that an individual with higher scores on curiosity will be more likely to express satisfaction in aspects of daily life and life in general [66]. Similar results were found by Ye et al. [34] concerning university life satisfaction. Therefore, these results indicate the validity of the use of the scale since the findings are compatible with the expected outcomes based on previous evidence from other authors [15,16,33,66].

In addition, these results are consistent with those obtained when using the Stretching and Embracing subscales instead of the total score. Thus, people with greater psychological wellbeing are also those who find the search for stimuli that can bring self-acceptance or personal growth most reinforcing [32]. Likewise, they perceive novel situations as a challenge rather than a threat, due to their ability to manage the tension that may be generated by the uncertainty of such situations [33]. Further, those who adopt successful strategies for exploring complex and challenging situations experience greater satisfaction [66].

Nevertheless, taking into account the high correlation between the two subscales (r = 0.73) and the similarity between their correlations with the external variables measured in this study, in addition to the unidimensional structure obtained in the previous analyses, we do not see how, on the basis of these data, the use of two subscales can be justified.

Despite the contributions of this study, there are several limitations that warrant attention and discussion. On the one hand, the findings of this study are based on a sample of Spanish undergraduate students. The generalisability of these findings to other populations, such as children, secondary school students, or working adults is not ensured. Therefore, further studies based on these populations are necessary. On the other hand, due to the health research orientation of the present work, personality traits were not considered as constructs and variables used to assess the instrument’s content validity, as previous studies did. Future validation studies should include such variables to be compared with previous works. Finally, the variables used are constructs related to a good psychological adjustment and a good use of the resources available to achieve better health. However, the correlations between these variables and the CEI-II score were small, and no causal inference could be drawn. For that reason, further studies should focus on the potential association of curiosity with risky health behaviours and healthy lifestyles.

Additionally, further longitudinal studies would be of value for observing if there are differences between curiosity as a state and a trend. Another useful aim for future studies could be to obtain evidence of validity based on the content of the Spanish translation used, following the approach adopted by Tariq and Batool [67] for the Urdu language in Pakistan.

## 5. Conclusions

In conclusion, the results of the analysis of the psychometric properties of the CEI-II have shown a high internal consistency for both the total scale and its subscales. However, the similarities and the strong correlation between sub-scales, along with the better fit shown by the most parsimonious model structure (the one-dimensional model), support the use of the scale’s total score instead of the two-dimensional structure proposed by the authors of the original version. Therefore, considering its limitations, the Spanish version of the CEI-II is recommended for investigating not only the relationship of curiosity with other personality traits but also how this construct is related to health-related behaviors and how to use it for effective health interventions in university students. In our view, this work contributes to the field of health research according to the bio-psychosocial model proposed by the International Conference on Health Promotion [1].

## Figures and Tables

**Figure 1 healthcare-11-01128-f001:**
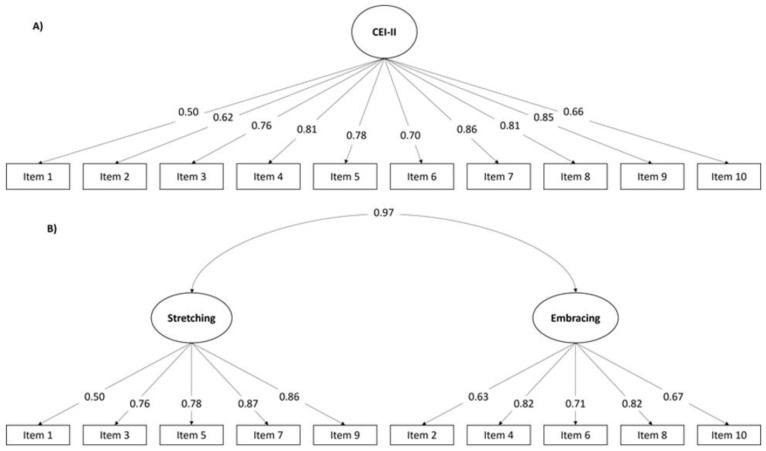
Confirmatory factor analysis path diagrams of the two models proposed: (**A**) One-dimensional model, and (**B**) Two-dimensional model. Standardised weights are presented.

**Table 1 healthcare-11-01128-t001:** Descriptive statistics of the CEI-II items.

Item	Mean	SD	Skew	Kurtosis
1. In novel situations I actively seek out as much information as I can	3.543	0.913	−0.331	−0.076
2. I’m the kind of person who really enjoys the uncertainty of everyday life	2.859	1.135	−0.124	−0.762
3. I’m very comfortable doing something complex or challenging	3.535	1.015	−0.384	−0.274
4. Wherever I go, I look for new things or experiences	3.6	0.962	−0.476	−0.067
5. I see challenging situations as an opportunity to grow and learn	3.818	0.895	−0.495	−0.112
6. I like to do things that are a little scary	3.19	1.134	−0.243	−0.731
7. I am always looking for experiences that challenge my thinking about myself and the world in general	3.322	1.087	−0.362	−0.480
8. I like jobs that are excitingly unpredictable	3.045	1.074	−0.184	−0.573
9. I often look for opportunities to challenge myself and grow as a person	3.565	1.01	−0.466	−0.216
10. I’m the kind of person who embraces unknown people, events, and places	3.036	1.235	−0.044	−0.953

**Table 2 healthcare-11-01128-t002:** Pattern matrix based on the polychoric matrix using unweighted least squares.

Item *	One Dimension			Two Dimensions			
	F1	h2	u2	F1	F2	h2	u2
CEI-II 1	0.466	0.217	0.783	0.569		0.254	0.746
CEI-II 2	0.519	0.270	0.730		0.506	0.307	0.693
CEI-II 3	0.775	0.601	0.399	0.884		0.677	0.323
CEI-II 4	0.746	0.557	0.443		0.406	0.560	0.440
CEI-II 5	0.802	0.643	0.357	0.879		0.710	0.290
CEI-II 6	0.637	0.405	0.595		0.492	0.428	0.572
CEI-II 7	0.741	0.549	0.451	0.512		0.550	0.450
CEI-II 8	0.735	0.540	0.460		0.828	0.660	0.340
CEI-II 9	0.758	0.575	0.425	0.510		0.575	0.425
CEI-II 10	0.550	0.303	0.697		0.638	0.377	0.623

* F = Number of factors’ model; h2 = communality of the item; u2 = uniqueness of the item; Weights lower than 0.40 are hidden.

**Table 3 healthcare-11-01128-t003:** Fit indexes for the model tested.

Model *	χ^2^	df	*p*-Value	CFI	TLI	RMSEA	90% CI RMSEA	SRMR
One-Dimension	86.853	35	<0.001	0.996	0.995	0.061	0.045–0.077	0.044
Two-Dimensions	83.929	34	<0.001	0.996	0.995	0.060	0.044–0.077	0.042

* χ^2^ = Chi-Square statistic; df = degrees of freedom; CFI = Comparative Fit Index; TLI = Tucker Lewis Index; RMSEA = root mean square error of approximation; SRMR = Standardized Root Mean Square Residual.

**Table 4 healthcare-11-01128-t004:** Goodness of fit indexes for the different steps of the factorial invariance analysis.

Model *	Variable	Invariance	χ^2^/df	RMSEA	∆ RMSEA	CFI	∇ CFI	TLI	SRMR
One-Dimension	Gender	Configural	2.051	0.051	-	0.995	-	0.995	0.047
		Metric	2.100	0.052	0.001	0.994	−0.001	0.995	0.049
		Scalar	2.318	0.057	0.005	0.993	−0.001	0.994	0.049
		Strict	2.380	0.059	0.002	0.991	−0.002	0.994	0.053
	Age Group	Configural	1.236	0.034	-	0.998	-	0.998	0.052
		Metric	1.269	0.037	0.003	0.997	−0.001	0.998	0.054
		Scalar	1.369	0.043	0.006	0.996	−0.001	0.997	0.054
		Strict	1.578	0.054	0.011	0.993	−0.003	0.995	0.064
Two-Dimensions	Gender	Configural	1.787	0.044	-	0.996	-	0.996	0.042
		Metric	1.772	0.044	0	0.996	0	0.996	0.043
		Scalar	2.057	0.051	0.007	0.994	−0.002	0.995	0.043
		Strict	2.026	0.050	−0.001	0.994	0	0.995	0.045
	Age Group	Configural	1.054	0.016	-	1	-	1	0.046
		Metric	1.086	0.021	0.005	0.999	−0.001	0.999	0.048
		Scalar	1.159	0.028	0.007	0.998	−0.001	0.999	0.048
		Strict	1.340	0.041	0.013	0.996	−0.002	0.997	0.058

* χ^2^/df = Chi-square statistic/degrees of freedom; Conf = configural invariance; Metric = metric invariance; Scalar = strong invariance; Strict = strict invariance.

**Table 5 healthcare-11-01128-t005:** Internal consistency coefficient by gender and age group.

Coefficient *	Gender		Age Group			
	Male	Female	18–19	20–21	22–23	24–26
CEI-II α	0.91	0.90	0.90	0.90	0.90	0.93
Stretching α	0.89	0.84	0.85	0.85	0.86	0.90
Embracing α	0.81	0.83	0.81	0.85	0.77	0.86
CEI-II ω	0.89	0.89	0.88	0.89	0.88	0.92
Stretching ω	0.86	0.82	0.82	0.84	0.83	0.89
Embracing ω	0.78	0.80	0.79	0.82	0.74	0.84
CEI-II Guttman	0.91	0.90	0.90	0.90	0.90	0.93
Stretching Guttman	0.84	0.78	0.79	0.79	0.79	0.83
Embracing Guttman	0.76	0.78	0.77	0.80	0.73	0.81

* α = Ordinal Alpha; ω = McDonald’s Omega.

**Table 6 healthcare-11-01128-t006:** Correlations between CEI-II scores (scale and subscales) and other variables.

Variable	Stretching	Embracing	GHQ-12	SWLS	SOC-13
CEI-II	0.92 *** [0.91, 0.93]	0.94 *** [0.93, 0.94]	−0.21 *** [−0.27, −0.14]	0.27 *** [0.20, 0.33]	0.21 *** [0.14, 0.27]
Stretching		0.73 *** [0.70, 0.76]	−0.21 *** [−0.28, −0.15]	0.26 *** [0.20, 0.33]	0.20 *** [0.13, 0.27]
Embracing			−0.18 *** [−0.24, −0.11]	0.24 *** [0.18, 0.31]	0.19 *** [0.12, 0.26]
GHQ-12				−0.45 *** [−0.50, −0.39]	−0.63 *** [−0.67, −0.59]
SWLS					0.49 *** [0.44, 0.54]

Note: *** *p* < 0.001.

## Data Availability

The data that support the findings of this study are available from the corresponding author upon reasonable request.

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
