# Peer review of "Validation of the Curiosity and Exploration Inventory-II in Spanish University Students"

_healthcare, 2023, doi:10.3390/healthcare11081128_

Round 1
Reviewer 1 Report
Thank you for the opportunity to review this paper for the submission to the HealthCare Journal. I think that the authors have proposed a new version of the Curiosity and Exploration Inventory-II in Spanish university context, but is not very clear what is the better factorial version of this inventory in this study context.
The authors have dedicated a very brief introduction to this paper, reducing the attention to the different topics (life satisfaction, sense of coherence, and psychological wellbeing). I believe that it is a mistake.
For statistical analyses, the authors tested significant models in relation to the factorial analysis due for the Curiosity and Exploration Inventory-II .
Another inadequate part of this paper is present in conclusions. These are very limited to address the future research in this field.
Reviewer 2 Report
Thank you for the opportunity to review the manuscript entitled, “Validation of the curiosity and exploration inventory-II in Spanish university students. ”
Below I provide my overall impressions followed by more specific comments.
OVERALL COMMENTS:
1. This manuscript draws attention to an important topic in the psychometric validation of the Curiosity and Exploration Inventory-II among Spanish undergraduate.
2. Parts of this manuscript could be strengthened. For example, include more discussion on supports for both one factor and two factor solution of CEI-II in Spanish undergraduates. Not just statistical supports but more discussion in content areas.
3. Please provide more detailed information regarding participants. Other than gender, age, and major, any socio-demographic information was collected?
4. Please provide more detailed information regarding the sample selection process and exclusion criteria, if any. When was data collection performed and completed?
5. In Tables 3 through 5, you have divided total study sample to four age groups. Is it necessary to divide sample based on two-year interval? Did you expect any invariant discrepancy across these four groups? If so, in what sense? Please add an explanation for this subsampling.
6. Minor correction needed: typo in line 409.
Reviewer 3 Report
For paragraphs 1 and 2.1 to 2.3, I have no notes (except for further comments)
Lines 176-181 - From the text it is understood that the polychoric correlation matrix was used only for the factor analyzes (EFA and CFA) and for the reliability analysis, but not for all the other analyses. The use of polychoric correlations implies that the variables are not all metrics, but that some variables are qualitative.
However, on lines 203-205, it is stated that Pearson's correlations have been used for concurrent validity. Why?
Lines 210-213 - It is stated that CEI-II is not normally distributed, but there are no indications on how this problem has been addressed: polychoric correlations?, robust analyses?, bootstrappig? Maybe it was ignored?
Lines 236 – The CFA is used to solve the “one-dimension and two-dimension models” dilemma but then there is no decision. Indeed, in line 251, it is assumed that both models are good and, subsequently, the authors work with both. Maybe it's worth replacing “a CFA was needed to decide” with something like “a CFA may be usefull to decide”.
Lines 243-244 and Figure 1 - Considering that the CFA on the two-dimensional model has a very high correlation between "Stretching" and "Embracing" (r=.97), perhaps it would be worth choosing the one-dimensional model. In fact, the two factors seem to measure the same thing.
Lines 261-281-As indicated above, is there a reason why the authors calculate and report both Alpha and Omega?
Rows 288-293 and Table 6 – As already indicated, the Pearson correlation is calculated for external or concurrent validity. I remind the authors that the probability associated with the correlations depends on df=N-2 (therefore already with N=402 all correlations higher than 0.12 are significant, p<0.001.). Perhaps it is better to use correlations as effect size (in r-family mode).
Thus, large (>.50) effect size for correlations within CEI-II; within GHQ, SWLS and SOC13; between small (>=.10) and medium (<=0.30) considering CEI-II (including Stretching and Embracing ) and the concurrent variables.
4. Discussion
This section solves a few of the above problems. The way the comments are presented suggests that the data analyzes were done "a priori" and only after that the authors tried to justify what had been done.
Wouldn't it be a good idea to resolve certain situations in advance and leave the discussions on the results here?
In particular, the fact that GHQ correlates negatively is indicated twice (lines 292 and 374) as something peculiar and unexpected. Anyone familiar with the GHQ knows that it does not measure well-being but malaise (high scores indicate "distress"). This information should be indicated in the paragraph "Psychological wellbeing" (lines 140-144).
5.Conclusions
The first part of the conclusions (lines 418-423) seem more than acceptable to me, while about lines 423-425, I am perplexed by the suggestion to use CEI-II "for investigating health related exploratory behaviours".
Author Response
Please see the attachment, thank you!
